# A Development of Visualization Technology through AR-Based Design Checklist Connection

**Hyejin Park and Seungyeon Choo \***

School of Architecture, Kyungpook National University, 80, Daehak-ro, Buk-gu, Daegu 41566, Korea; phj8598@knu.ac.kr
**\*** Correspondence: choo@knu.ac.kr

**Abstract:** Since the initial design review has the effect of minimizing the design changes needed in the later stages of an architectural project, a process of collaboration between architects, clients, and engineers in this design review is very important. Recently, design review using 3D models rendered in VR or AR, going beyond simple simulation, has been addressed in many studies. However, a synchronization function that provides immediate visualization of design changes is the focus, which has limitations in its ability to review factors required by law related to safety and design. The purpose of this study was, therefore, to develop AR-based design checklist linkage and visualization review technology that facilitates decision making by clients and architects during design review. For this purpose, a method for linking a design checklist in an AR environment using a game engine (Unity 3D) is proposed and various design review visualization functions are developed with consideration paid to the user interface. The efficiency of the design checklist linkage technology and visualization function developed in this study was verified in the pilot project.

**Keywords:** instructions; design review; decision making; augmented reality; design checklist; multi-dimension; visualization; mobile app

## 1. Introduction

### 1.1. Background and Purpose

The initial design review in an architectural project has the effect of reducing design changes needed in the later stages of said project by 5–40% [1], and as such, a process of collaboration between architects, clients, and engineers for such a review is very important [2]. Since BIM allows us to check and correct errors [3] in design immediately while viewing the 3D model, the introduction of this technology has greatly facilitated collaboration [4]. In practice, many design reviews (such as interference checks by construction type [5] and 4D simulations [6]) are performed by applying BIM, to improve design quality and performance. However, since the BIM design model displayed on a computer monitor is the one used in much of the design review process, there are limitations in its ability to produce a spatial sense, which often impacts the sense of reality that people have of it [7].

Recently, design reviews using 3D models designed in VR or AR, beyond simple simulation, have been addressed in many studies [8,9]. Among them, the metaverse, a higher concept of VR and AR, is an expanded virtual reality and is becoming an ever more important issue in the deep tech realm [10]. The metaverse is not simply a 3D space but a space where the intersection of the real and virtual worlds is reconstructed and fused using 3D technology [11]. Microsoft have built a data center to run a virtual reality space and launched a 'Data Center Virtual Experience Program' that is accessible by everyone [12]. In Korea, a 'Megacity' based on BIM technology was implemented in a virtual space, and a commercial market for digital real estate market was formed. In addition, 'Earth 2′, which was reproduced from Google Earth satellite maps and imagery, was implemented as a virtual real estate platform, in which even transactions are possible [13]. As such, VR and

AR are effective methods for an intuitive experience of a design proposal because they enhance the user's immersion and sense of reality. Twinmotion, a commonly used software package for rendering, provides via VR the ability to view 3D models designed through game engines in virtual reality. Fuzor also provides a function that allows users to feel realism through avatars using VR and AR [14]. However, the VR and AR functions in currently developed rendering software focus on 3D model exporting [15]. In design review many items including not only design (e.g., shape, material, and scale) but also legislation (e.g., scale, mass) and safety (e.g., barrier-free) should be considered comprehensively. For effective collaboration, the items of law, safety, and design elements considered in design review should be systematized and an interface in which users can review while viewing the design checklist in real time should be established.

The purpose of this study is, therefore, to develop AR-based design checklist linkage and visualization review technology that facilitates decision making by clients and architects during a design review. For this purpose, a method for linking a design checklist in an AR environment using a game engine (Unity 3D) is proposed and various design review visualization functions are developed with consideration paid to the user interface.

### 1.2. Method and Procedure

The methodological procedures of the study are shown in Figure 1.

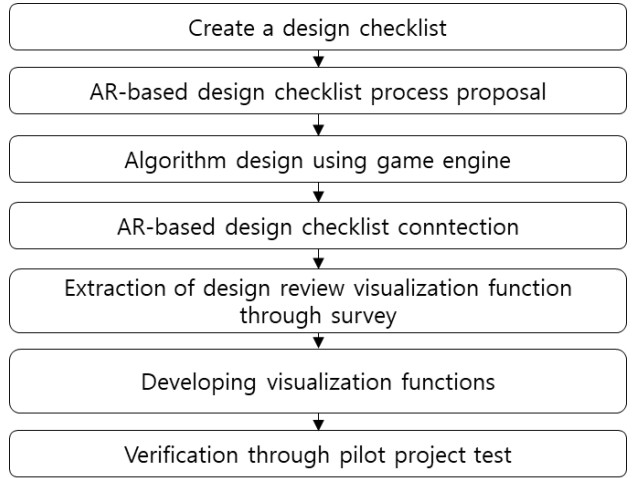

**Figure 1.** Method and procedure.

First, the knowledge of architects and the design checklists of design offices were investigated in order to generate a checklist of elements to be considered during the design process, through which a refined design checklist was derived based on common issues. Second, a process was presented to link the design checklist in the AR environment. For implementation, a script was written using a game engine and implemented as an algorithm. Third, for the development of AR-based design review visualization technology, essential functions were derived through a survey. Fourth, the visualization function mentioned above was developed using a game engine. Lastly, for verification of the AR-based design checklist linkage and the developed visualization function, a test was conducted through a pilot project using a mobile app, through which the practicality of use was verified.

## 2. Related Research

### 2.1. Case Study of Mixed Reality Technology in Architectural Design

Recently, technologies that mix real and virtual spaces have been applied to various industries and are becoming more prominent in the realm of deep tech [16]. The COVID-19 pandemic and the emergence of the digitally accustomed MZ generation (a term used in Korean discourse to refer to both Millennials and Generation Z together as a single cohort) is rapidly shifting social contact from the real world to the virtual one. Social

distancing is pushing people more and more people into online spaces. Communication, which was formed only in human relationships, has recently become possible in a virtual online environment called the Metaverse going beyond MR (mixed reality). The number of users of Roblox, a most famous metaverse platform, increased significantly from 12 million in 2018 to 42.1 million in 2021 [17], and mainly in the entertainment industry and game industry, related technologies are growing rapidly.

The construction industry consists of collaboration with a variety of people, from planning and construction to sales. The construction industry requires a lot of communication, but there is a limit to face-to-face communication. In order to solve this problem, MR technology has recently started to be applied (Figure 2).

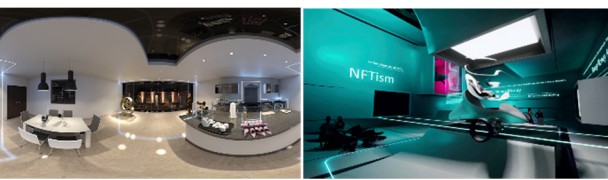

**Figure 2.** Cases study of NVIDIA Iray and NFTism.

NVIDIA, when designing their new Silicon Valley office building, developed the 'NVIDIA Iray' technology for users to experience the inside and outside of a building using VR. This technology, making use of ray tracing by calculating and applying the way in which individual rays of light behave, allows users to experience a realistic environment. In addition, in order to increase cooperation between departments in flow design, VR was used to increase the number of times employees encounter other departments. In the actual design process, more than 90% of the design was created through VR simulation [18]. Zaha Hadid, a famous architecture firm, implemented NFTism, a fictional art gallery, using the metaverse. A multiplayer online game was provided so that users could indirectly experience the design created through NFTism. A new community and society formed through communication with people who were wanted within the gallery served as an opportunity to form an economic infrastructure [19]. Since work in the construction industry, due to its nature, requires constant collaboration with various actors, research that combines VR and AR is likely to significantly contribute to this field. As a response to the COVID-19 pandemic, the construction environment is also changing centered on an Non-Contact one and it is urgent to develop an interface that enables immediate communication through VR. This study proposes and develops a way to link the design checklist to enable various forms of communication between the architect and the client in an AR environment during the design review and develops it through a game-based algorithm.

## 2.2. Research Trends Related to AR

The virtual aspects of AR have great potential for pre-construction stages, as AR can imitate risky settings that are difficult to simulate in real construction sites [20]. To effectively use AR technology for end-user involved design collaboration, it is necessary to ensure the effectiveness of the AR system from the end user's perspective. Moisés David Osorto Carrasco (2021) analyzed the effect on the design review effect through a survey using the holographic function of Microsoft HoloLens. In this study, it was estimated that MR-based design review could increase comprehension by more than 15% over 2D drawing-based methods [21]. Recently, BIM technology has been actively used to facilitate collaboration in the construction field [22]. James Garbett et al., (2021) developed a proposed framework for a collaborative BIM–AR system that provides the ability for users to interact in a synchronous environment [23]. Most efforts have mainly focused on technological development, and as such, limited attention has been paid to the end user's application of the AR system. The previous studies have shown AR to be an effective way of reviewing designed building models [24]. Jin Gang Lee et al., (2020) investigated the visual expression quality and user acceptance of architectural design using 2D monitors, VR,

and AR for 76 students. This study aimed to measure whether a design can be effectively expressed through a visualization platform by selecting design review elements (color, texture, shape, windows, nature, and scale) [7]. There are only a few prior studies related to the research on necessary functions and algorithms for improving AR-based usability during design review. In a design project, how architects, clients, and engineers actually collaborate to review a design and make a decision is a very important issue at present. This study intends to develop a methodology that can utilize BIM–AR for design review using a game engine and a visualization function considering UI/UX.

## 3. Design Checklist

In general, the architectural design process is divided into the following stages: pre-design, schematic design, drawing development, and working drawing. In the working drawing stage, since actual drawings for construction are prepared according to previous decisions, consultation is mainly made in the schematic design and drawing development stages [25]. Therefore, the scope of 'design review' in this study is limited to the schematic design to drawing development stages. The items in a design review vary depending on the knowledge and experience of the architect, the know-how of the design office, and the scope, scale, and type of building or buildings involved. For the generalization of the checklist elements to be considered during design, a preliminary survey was conducted for each design office. The design checklist was organized systematically by including qualitative and quantitative items around common key elements. What was found was that the elements requiring decision making in the design stage were largely design and safety related. The design-related items numbered 51, categorized into plan review ($n = 25$), scale review ($n = 12$), and material review ($n = 14$), while the safety-related ones numbered 14, categorized into fire-related review ($n = 5$), evacuation-related review ($n = 2$), and obstacle-related review items ($n = 7$). Figure 3 and Table 1 show this visually. Table S1 contains data for the checklist.

Design checklists were coded so that items to be reviewed for each object or place were linked through a pop-up function while viewing the BIM model in the AR environment. The codes were derived from the location names, typically abbreviations featuring the first letter of the location name, with additional letters to avoid repeats. Table 2 shows a complete list of the codes.

**Table 1.** Example items for a design checklist.

| | Design Review Items | | Safety Review Items |
|---|---|---|---|
| Plan | Can you see the elevator at the main entrance? Does it connect from the top floor of the stair hall to the outside roof? etc. | Fire | Is the window open to the outside in case of fire? Is the direction and method of opening and closing the window appropriate? etc. |
| Scale | Is the size of the room area appropriate? Is the height/height of each floor appropriate? etc. | Refuge | Are the non-slip pads of the evacuation stair or special evacuation stairwells in conspicuous colors? Is the effective width of the passageway for evacuation more than 120 cm? etc. |
| Material | Is the approximate interior material (wall, floor, ceiling) appropriate? Is the window glass color appropriate? etc. | Barrier free | Are mechanical devices, such as escalators or moving walk, warning signs for safety (risk) at the bottom of the entry? Is there an auxiliary handle on the handrail of the public corridor of the facility? etc. |

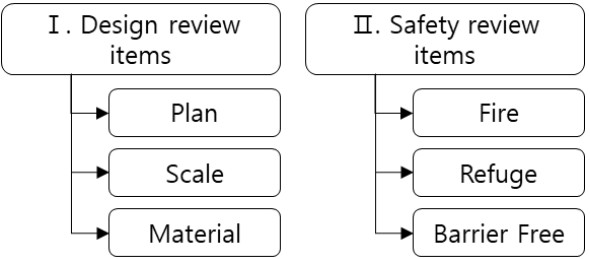

**Figure 3.** Design checklist items.

**Table 2.** Codes for design review items.

| Code | Location | Code | Location |
|------|----------|------|----------|
| E | Entrance | T | Toilet |
| H | Hall | S | Stair |
| HW | Hallway | SH | Stair hall |
| F | Floor | ES | Evacuation stair |
| FE | Floor of entry | SES | Special evacuation stair |
| W | Wall | EV | Elevator |
| C | Celling | MW | Moving walk |
| R | Room | WD | Window |
| LR | Living room | D | Door |
| K | Kitchen | SK | Sunken |
| B | Bathroom | RA | Rail |

When there was more than one matter to be reviewed, they were presented in the form of 'code-numbers' such as F-001, F-002, F-00N to avoid duplication. This allows architects to distinguish between and use different design checklists. The current practice of design review has limitations in satisfying the exact design desired by the client since it focuses on legal matters. The design checklist presented in this study is centered on qualitative items desired by the client, such as design, scale, and material, differentiating it from conventional checklists used in current practice. In the BIM model, the code was input on the left side of the field indicating 'Type' (Figure 4). This allows the code to be practically used in object recognition when implementing a design checklist as a visualization in a game engine.

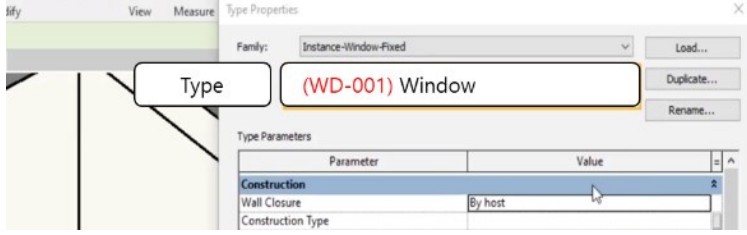

**Figure 4.** Method to input check code in the BIM model.

## 4. Methodology

### 4.1. Process of AR-Based Design Checklist Connection

To support decision making by various participants in a design review, we propose a method to link the BIM model with an AR-based design checklist. Based on the design checklist derived in Section 3, a method to input the code into the BIM model was devised. Figure 5 shows the development process.

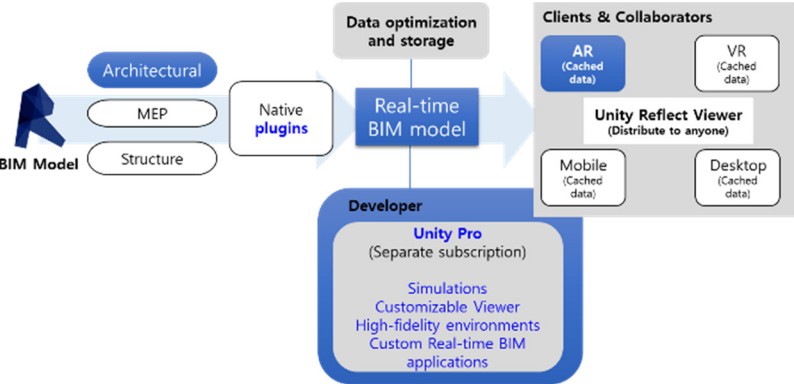

**Figure 5.** Process of AR-based design checklist connection.

A game engine was used to combine the BIM model and the checklist in the AR-based environment, and a script was written in the commonly used Unity 3D Pro; functions were developed through customization. First, BIM-based 3D modeling was performed and only the architectural model was modeled. The model was exported to the game engine through use of a plug-in. The BIM model can be opened in the game engine, which runs in the cloud and the design review checklist is linked through a C#-based script [26].

### 4.2. Algorithm Design and Implementation

Figure 6 shows a plan for linkage with the AR-based design checklist. The visual direction in which the design checklist should appear in the spatial location where the avatar or person is in the AR environment is indicated in the drawing.

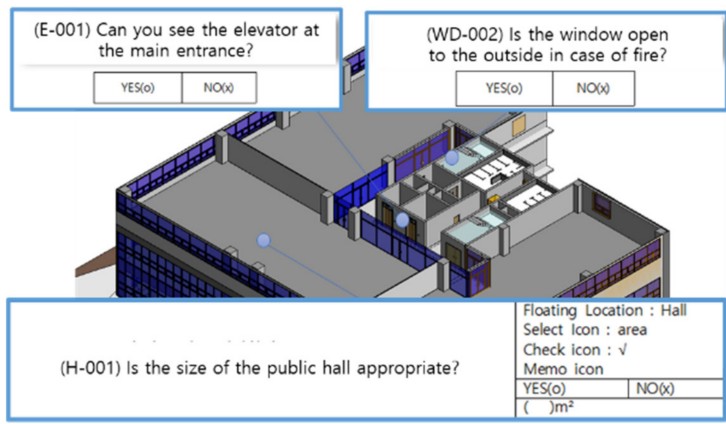

**Figure 6.** Example of design checklist connection.

Figure 7 shows the meaning of the symbols in the algorithm. An algorithm was designed, as shown in Figure 8, to link with the design checklist.

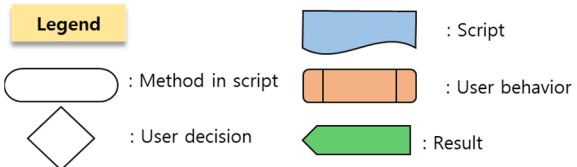

**Figure 7.** Meaning of Symbols for Algorithm.

First, an algorithm that generates one toggle button for one item to be reviewed in the AR environment was created. When the toggle button is selected (On Project Chosen), the corresponding checklist (RegulationDB) floats above it (Readcsv). When an input value is given to 'find a code with a specific regulation' (Search Code) in the DB of the design

checklist, the system is designed to display a toggle button (Toggle shown) if such a code exists in the database (Toggle check). When 'Yes' is selected, the toggle disappears, and when 'No' is selected, the toggle button changes to a caution symbol. In addition, the algorithm was created so that the result of selecting 'Yes' or 'No' is saved in a file.

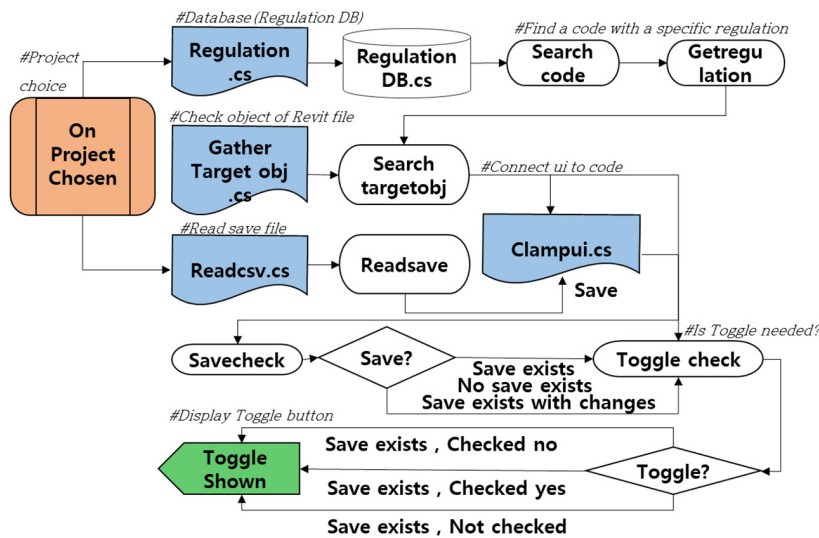

**Figure 8.** Algorithm to connect design checklist.

- **Toggle function**

    In this study, a toggle function is implemented so that the object and its location is displayed for the design review. After the toggle is activated, if there is a design review related to the location and object, that toggle disappears if there is no problem but remains if there is a problem (Figure 9).

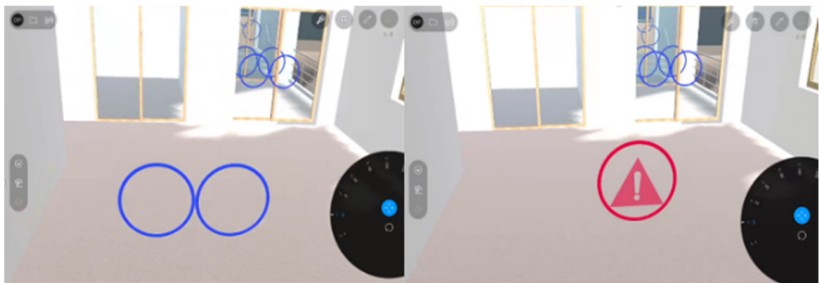

**Figure 9.** Implementation of toggling function.

- **Floating & checking functions**

    A floating function was implemented that allows items and contents related to the corresponding design review items to be displayed on the user's screen in conjunction with the checking technology (Figure 10).

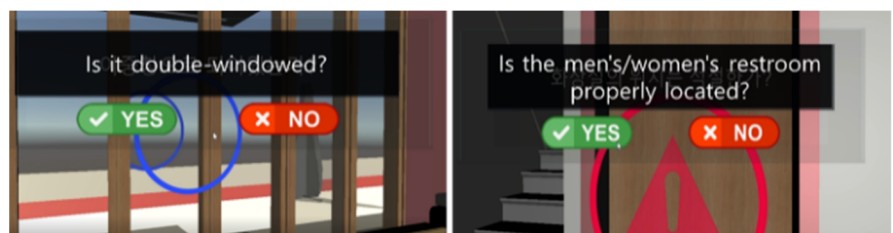

**Figure 10.** Implementation of floating and checking function.

## 5. Development of Visualization Technology

### 5.1. Extract Function

Based on the design checklist made in Section 3, the development scope and functions of AR-based design review visualization were determined. A questionnaire survey was conducted with 20 design-related workers. It investigated what kind of visualization review function would be required with the assumption that a design review is conducted focusing on the items addressed in the checklist. The necessary functions according to the checklist were sorted into six categories: location function, scale and area function, object function, dimension measurement function, user function, and light function; for example, as shown in Table 3 for users to select (multiple check allowed) the functions necessary for review of the given item. Table S2 is full view of the survey results.

**Table 3.** Example of questionnaire items to determine development scope and functions of AR-based design review visualization.

| Checkpoint & Discussed Items | Site View | | Size / Area View | | Object View | | Dimension Measurement | | | User Review | | Light Review | |
|---|---|---|---|---|---|---|---|---|---|---|---|---|---|
| | Site | Plan | Object | Scale | Area | Type | Texture | Area | Scale | Size | Usability | Foot Traffic | Natural Light | Light |
| Orientation of building | | √ | | | | | | | | | | | √ | √ |
| Countermeasure for ground level difference | √ | √ | | | | | | √ | | √ | | | | |

Questionnaire survey scores for 20 subjects were summed up by function and a function with higher score was determined to be more urgently developed. The survey results showed that the dimension measurement function (908), the usability function (724), and the material review function (644) had relatively higher scores, indicating that development was necessary, and that the function for light review had a relatively lower score, indicating low usability (Table 4).

**Table 4.** Survey results for function development of visualization review functions.

| Checkpoint | | Score |
|---|---|---|
| Site review | Site | 164 |
| | Plan | 384 |
| | Object | 420 |
| Size/area review | Scale | 640 |
| | Area | 492 |
| Object review | Type | 608 |
| | Texture | 644 |
| Dimension measurement | Area | 544 |
| | Scale | 532 |
| | Size | 908 |
| User review | Usability | 724 |
| | Foot traffic | 600 |
| Light review | Natural light | 240 |
| | Light | 312 |
| **Total** | | **7212** |

In Section 5.2, the AR-based visualization function was developed based on the identified functions by creating an algorithm using a game engine.

### 5.2. Development of Visualization Technology

- **Measurement function**

In this study, a measurement function was developed to measure the dimensions of objects and spaces. In consideration of the user-based interface, an algorithm was created so that the user can employ the measurement function when he/she clicks a simple icon (Figure 11).

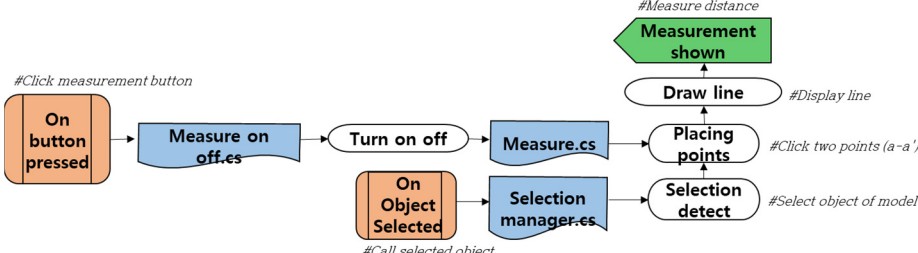

**Figure 11.** Algorithm to develop measurement function.

An algorithm was designed, for the measurement of distance using the measurement button (On button pressed), to display a line (Draw line) when starting point and other point are selected (Placing point). To increase measurement accuracy, clicking the measurement button activates the vertices of all objects. If a user selects two points via the snap method, the corresponding length is displayed in millimeters (Measurement shown) (Figure 12).

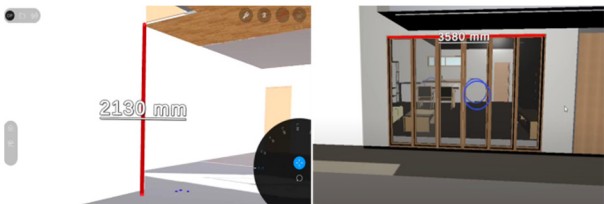

**Figure 12.** Implementation of measurement function.

- **Material change function**

The material change function was developed using the material library provided by BIM. The material library is based on the Revit library and contains 1849 material source files. The material change algorithm proceeds in the following order: (1) material selection, (2) related material list pop-up, (3) RGB color change. Figure 13 shows the development algorithm.

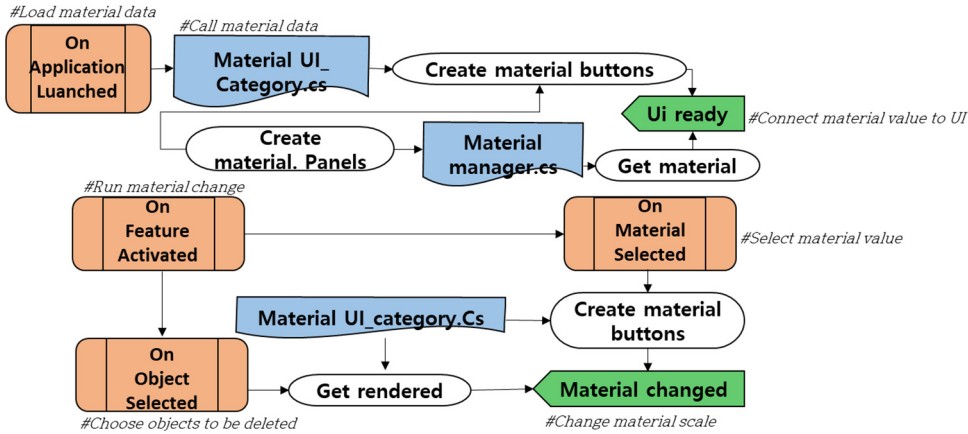

**Figure 13.** Algorithm for development of material change function.

After the material images (Material UI category) are imported (Get material) into the game engine, clicking the change material icon (On feature activated) leads to visualization of the various material lists (Figure 14). For adjustment of the material image scale, a material scale change algorithm (Material changed) was created to adjust the X and Y values of each image [27].

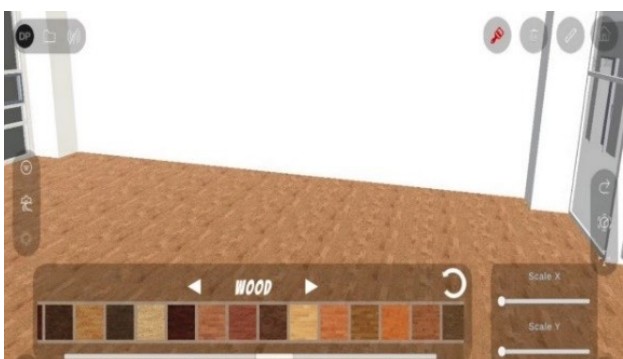

**Figure 14.** Implementation of material change function.

- **Reporting function**

As shown in Figure 15, an algorithm was created to develop the function to be reported through the design review checklist.

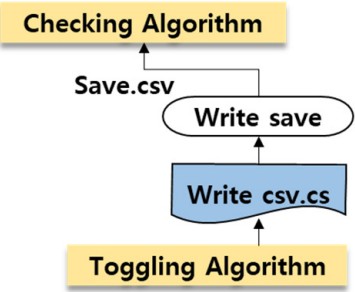

**Figure 15.** Algorithm for development of report function.

The algorithm was developed so that the outcome values of true or false, which are the checked result in the previously developed checking algorithm, is given (Save) through integrated (Write csv) Excel (Figure 16).

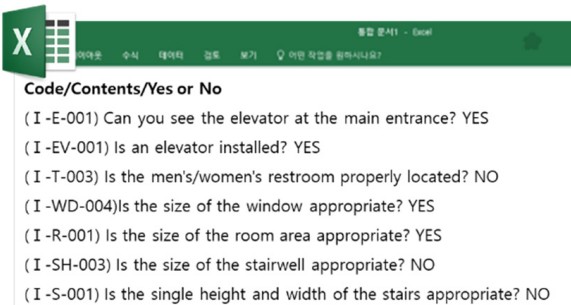

**Figure 16.** Results extracted to excel through reporting.

*5.3. Mobile App Implementation*

Here, the functions developed to be easily accessed by the user are verified through the mobile app. An .apk file was extracted from the build in the game engine and downloaded to a mobile device. For the implementation using AR, the method of uploading to the cloud

and running a BIM model was used in verification. Figure 17 shows the interface of the mobile app developed in this study.

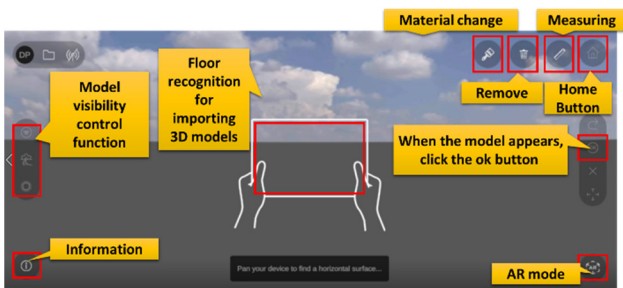

**Figure 17.** Mobile app interface.

In AR mode, the scale can be adjusted from 1:1 to 1:10,000; in this study, as shown in Figure 18, floor detection is set to lead to the displaying of the BIM model on the screen.

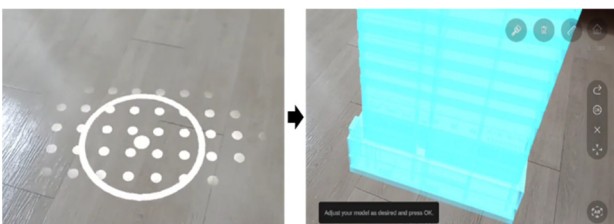

**Figure 18.** BIM model displayed on screen through floor detection.

In order to verify the practical utility, a pilot project was conducted and the previously developed checklist linkage functions (checking and floating, measuring, material changing, and reporting) were tested. The test results showed that a checklist toggle button should be created in the center of the object, and that clicking the toggle button should lead to the display of an item in the checklist. In addition, it was found that the design review results, through the mobile app, should be automatically saved as a .txt file in the app folder. The measurement function showed high accuracy only at the 1:1 scale. It was found that there is a need for future improvement so that the dimensional value can be changed flexibly according to any change in scale. An experiment was conducted to identify a more effective BIM model for material change. In the model created by separating architecture and structure, internal and external objects were recognized separately, and materials could be applied to both separately. However, the BIM model in which the material is divided into several layers in one object failed at recognizing the inside and outside as separate surfaces, and the whole model was changed to a single material. This shows that the separation of architecture and structure in the BIM model development facilitates material changes in the AR environment.

## 6. Conclusions

The results are as follows:

First, a questionnaire survey was conducted targeting the architecture office to generalize checklist elements to be considered during design and this allowed us to systemize the design checklist focusing on the common items. Second, a methodology for linking the design checklist with AR was proposed. Each algorithm for toggling, checking and floating was developed to link the design checklist using the game engine. The design checklist was a pop-up on the object and can be linked automatically in the form of AR through the code entered in the object of the BIM-based model. Third, a questionnaire survey was performed to derive the scope and functions of an AR-based design review visualization and the results showed that the Measurement function, Material Review function, and Usability Review function should be developed first in the design review. Fourth, an algorithm was

created in a game engine to develop a design review visualization function. The measurement function, material change function, and report function were developed through user-based interface design and verified through AR implementation. Fifth, the efficiency of the design checklist linkage technology and visualization function developed in this study was verified in the pilot project. The interface was reviewed through a mobile app and the operability of the developed functions was verified. Uploading the BIM model with the check code to the cloud environment showed that the functions developed in this study worked normally. This proves that the developed functions can be implemented only by inserting the check code into the BIM model. Therefore, the utility value of decision-making support during architectural design review was demonstrated.

We plan to enhance the visual design review elements by utilizing a library database that includes lighting and environmental elements. This is a convergence study of BIM and AR and is expected to be used as a basis for the development of multidimensional visualization technologies applied in design review in the future.

**Supplementary Materials:** The following supporting information can be downloaded at: https://www.mdpi.com/article/10.3390/app12126126/s1, Table S1: Example items for a design checklist; Table S2: Survey results.

**Author Contributions:** H.P., conceived experiments, analyzed data, wrote papers, investigated prior research, and edited thesis; S.C., supervised the research. All authors have read and agreed to the published version of the manuscript.

**Funding:** This research is a basic research project in the field of science and technology that was conducted with the support of the Korea Research Foundation with funding from the government (Future Creation Science) in 2022. Assignment number: 2019R1A2C2006983. This research is a basic research project in the field of Ph.D. student research incentive project that was conducted with the support of the Korea Research Foundation with funding from the government (Future Creation Science) in 2021. Assignment number: NRF-2021R1A6A3A13045691.

**Institutional Review Board Statement:** Not applicable.

**Informed Consent Statement:** Not applicable.

**Data Availability Statement:** Not applicable.

**Conflicts of Interest:** The authors declare no conflict of interest.

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
