# Peer review of "A Development of Visualization Technology through AR-Based Design Checklist Connection"

_applsci, doi:10.3390/app12126126_

Round 1
Reviewer 1 Report
This paper presents an AR-based design checklist linkage and visualization to facilitate decision-making. It is well-implemented work. Some minor issues include proofreading the manuscript, some claims need literature to backup (The introduction part). Some related work in the introduction part should be moved to the related work part. Personally speaking, a high-level research gap should be identified in the introduction part by e.g., summarizing the pat work into several categories instead of listing them one by one.
Author Response
We appreciate much for your helpful review comments. It gave us an opportunity to enhance our paper.
We revised and supplemented the paper based on the reviews as follows.
Please see the attachment.
Thanks you.

Reviewer 2 Report
The authors try to introduce the novel concept, but the novelty of the article is not enough. Anyhow there is a major need to revise the work with the following comments.
1. Although the authors have described the motivation and contribution of the proposed research, yet it is not enough to justify it.
2. No coding or programming is described in the manuscript to justify the results.
3. Table 3. is not explained properly, from where the data is collected. If it is collected from a survey then there should be its proof or justification?
4. Is the whole article consists of abstract theory or a survey? No numerical analysis or numerical data is used?
I suggest summarizing the conclusions section briefly, to focus on the results you get, the method you propose, and their significance.
Author Response

(The authors gave the same response as above.)

Round 2
Reviewer 2 Report
NO